# Profiling miRNAs of Teleost Fish in Responses to Environmental Stress: A Review

**DOI:** 10.3390/biology12030388

**Published:** 2023-02-28

**Authors:** Quanquan Cao, Hailong Zhang, Tong Li, Lingjie He, Jiali Zong, Hongying Shan, Lishi Huang, Yupeng Zhang, Haifeng Liu, Jun Jiang

**Affiliations:** 1College of Animal Science and Technology, Sichuan Agricultural University, Chengdu 611130, China; 2MARBEC, University Montpellier, CNRS, IFREMER, IRD, 34090 Montpellier, France

**Keywords:** fish, environment stress, miRNA, environment factors, target mRNA

## Abstract

**Simple Summary:**

miRNAs are a class of endogenous and evolutionarily conserved noncoding short RNA molecules that post-transcriptionally regulate gene expression through sequence-specific interactions with the UTRs of mRNAs and are capable of controlling gene expression by binding to miRNA targets, thus inhibiting translation, degradation and the regulation of mRNA and interfering in the final protein output. The understanding of these small molecules has expanded significantly in environmental stress mechanisms in the past few years, while recent studies are common on the characteristics and functions of miRNAs in teleost. Environmental factors may influence the transcriptional and posttranscriptional regulators of miRNAs, contributing to nearly all biological processes. In this review, progress is reported on the profiling miRNAs of teleost fish in responses to environmental stress.

**Abstract:**

miRNAs are a class of endogenous and evolutionarily conserved noncoding short RNA molecules that post-transcriptionally regulate gene expression through sequence-specific interactions with mRNAs and are capable of controlling gene expression by binding to miRNA targets and interfering with the final protein output. The miRNAs of teleost were firstly reported in zebrafish development, but there are recent studies on the characteristics and functions of miRNAs in fish, especially when compared with mammals. Environmental factors including salinity, oxygen concentration, temperature, feed, pH, environmental chemicals and seawater metal elements may affect the transcriptional and posttranscriptional regulators of miRNAs, contributing to nearly all biological processes. The survival of aquatic fish is constantly challenged by the changes in these environmental factors. Environmental factors can influence miRNA expression, the functions of miRNAs and their target mRNAs. Progress of available information is reported on the environmental effects of the identified miRNAs, miRNA targets and the use of miRNAs in fish.

## 1. Introduction

miRNAs are a class of endogenous and evolutionarily conserved noncoding short RNA molecules (18–25 nucleotides) that post-transcriptionally regulate gene expression through sequence-specific interactions with the 3′-untranslated regions (UTRs) of mRNAs and are capable of controlling gene expression by binding to miRNA targets, thus inhibiting translation, degradation and the regulation of mRNA and interfering in the final protein output [1,2]. Some miRNA targets can lead to the degradation and translational repression of mRNA. Only when the 5’-end of the miRNA sequence is complementary to the 3′-UTR of the mRNA can the recognition and interaction between miRNAs and mRNAs take effect, making miRNA a global regulator accounting for spatiotemporal gene expression pattern [3,4].

miRNA was first described in the roundworm (*Caenorhabditis elegans*) 30 years ago [5]. Soon afterward, miRNAs were found in the genomes of plants, animals and viruses [5,6,7]. Initial studies focused on unmasking miRNA libraries and determining their function (regulation of metabolism, growth and homeostasis) in domestic animals such as poultry and fish [2]. The miRNAs of teleost were firstly reported in zebrafish development, but there are recent studies on the characteristics and functions of miRNAs in fish, especially on the characteristics of miRNAs in fish compared with mammals [8,9]. To date, several miRNA databases of fish species have been identified and annotated on miRBase (http://www.mirbase.org/index.shtml, accessed on 7 June 2018) based on genome, including 1593 miRNA entries for zebrafish (*Danio rerio*), 1507 miRNA entries for Nile tilapia (*Oreochromis niloticus*), 917 miRNA entries for Atlantic cod (*Gadus morhua*), 869 miRNA entries of Atlantic salmon (*Salmo salar*), 534 miRNA entries for Burton’s mouth-brooder (*Astatotilapia burtoni*), 486 miRNA entries for Channel catfish (*Ictalurus punctatus*), 440 miRNA entries fir Nyassa blue cichlid (*Metriaclima zebra*), 433 miRNA entries for Fairy cichlid (*Neolamprologus brichardi*), 432 miRNA entries for Victoria cichlid (*Pundamilia nyererei*), 314 miRNA entries for Japanese rice fish (*Oryzias latipes*), 280 miRNA entries for Common carp (*Cyprinus carpio*), 255 miRNA entries for Pufferfish (*Fugu rubripes*), 245 miRNA entries for Spotted green pufferfish (*Tetraodon nigroviridis*), 78 miRNA entries for Atlantic halibut (*Hippoglossus hippoglossus*), 58 miRNA entries for Olive flounder (*Paralichthys olivaceus*) and 54 miRNA entries for Electric eel (*Electrophorus electricus*) (Table 1).

Environmental factors including salinity, oxygen concentration, temperature, feed, pH, environmental chemicals and seawater metal elements may affect the transcriptional and posttranscriptional regulators of miRNAs, contributing to nearly all biological processes. Environmental factors are the apparent reason for specified stress type. The survival of aquatic fish is constantly challenged by the changes in these environmental factors. To our knowledge, no review has brought together information on the profiling miRNAs of teleost fish in response to environmental stress. Thus, we report on the progress of available information on environmental effects on the identified miRNAs and the use of miRNAs in fish (Table 2).

## 2. Salinity

miRNAs have been found to play a key role in regulating liquid and electrolyte balance [33]. Many studies have elucidated the significance of miRNAs in ion transport. In recent years, research progress on small noncoding RNAs (such as miRNAs in regulatory circuits) has clarified the physiological and pathophysiological functions of miRNAs in osmotic-stress response [10,34].

### 2.1. miRNA Expression in Response to Environmental Salinity Changes

In the gill, the expression level of miRNA-429 decreased under high osmotic stress. Studies have shown that osmotic stress transcription factor 1 (OSTF1) may be regulated by miRNA-429 [12]. In our research group, differential expression patterns of miRNAs including miRNA-122, miRNA-190b, miRNA-124, miRNA-1a and miRNA-206 were described in gills of marbled eels (*Anguilla marmorata*) adapted to different salinities (freshwater, brackish water and seawater) [34]. Regulatory roles of miRNAs show potential in osmoregulation when miRNAs are either significantly upregulated or downregulated in different salinities.

### 2.2. miRNA Functions and Its Target mRNAs under Specific Salinity Environments

In the ionocyte of zebrafish embryos, the highly conservative miRNA-8 family regulated the ion transporters’ trafficking and accurately controlled the ions transport by targeting and modulating the expression of Na^+^/H^+^ exchanger regulatory factor 1 (NHERF1) [11]. NHERF1 is the regulator of the apical transport of transmembrane ion transporters. These miRNAs modulate the expression of NHERF1 and play a role in regulating the activity of Na^+^/H^+^ exchange (NHE) (Figure 1). The miRNA-8 may improve the function of NHEs’ inhibition in cAMP-mediated stress, which can allow NHE to occur independently in order to protect cAMP elevations [11]. The inactivation of miRNA-8 prevents zebrafish from responding to osmoregulation. In the absence of miRNA-8 family members, differentiation defects were observed in ionic cells. However, significant effects were observed in ionic cell physiology, suggesting that these miRNAs may have cell-type-specific functions and that members of the miRNA-8 family may play a key role during cell development and differentiation [11]. Similarly, another miRNA, miRNA-200, has a significant effect on NHE, which is involved in epithelial ion transport to inhibit the expression of NHERF1 and impair the osmotic responses of cells (Figure 1). Meanwhile, increased osmotic sensitivity can lead to Na^+^ accumulation in ionic cells [11].

In the kidney of Nile tilapia, miRNA-30c, a novel regulator, can regulate osmotic balances, and the loss of its function will lead to disruption of the fish, making it unable to respond to osmoregulation [14].miRNA-30c responded to osmoregulation by targeting heat shock protein 70 (HSP70). The inhibitory effect of miRNA-30c was reported on renal HSP70 expression under hyper-osmoregulation [14]. miRNA-429 (another member of the miRNA-8 family) has a feedback regulation and is full of complexity in osmotic signaling networks in euryhaline fish [12]. The loss of miRNA-429 function substantially led to a significant increase in OSTF1 level, resulting in changes in ion concentration and ion balance related to osmotic sensory transduction [12]. Experimental validation confirmed the direct interaction of miRNA-429 andOSTF1, suggesting that miRNA-429 play a key role in regulating continuous transcriptional induction of OSTF1 under high osmotic stress in tilapia. Insulin growth factor 1 (IGF-1) has been identified as the target gene of miRNA-206 and IGF-1 treatment can up-regulate the expression of Na^+^, K^+^-ATPase (NKA) and Na^+^, K^+^, Cl^−^ cotransporter (NKCC) in tilapia [15] (Figure 1). PC-5P-27517 and 3P-50929 are two new miRNAs that play a role in mediating the expression of aquaporin 10 (AQP10) (Figure 1). AQP10 is an intact membrane protein of the aquaglyceroporin family and acts as a permeable channel in the epithelial cells of the tissue. The responses of PC-5P-27517 and 3P-50929 to heat stress were up-regulated in genetically modified tilapia. The expressions of target gene AQP10 were inhibited, indicating that the transformation of water and glycerol was reduced in the liver during heat stress and was impaired in cell membrane fluidity [26].

In the Atlantic killifish (*Fundulus heteroclitus*), several potassium voltage-gated channel homologues are the targets of miRNA-135b. The response of miRNA-135b to arsenic-containing saline is inhibited under combined exposure conditions. The individual target could be the critical mediator in high osmotic responses, helping to explain how arsenic reduces osmotic tolerances. The discovery of significant interactions was found between arsenic and salinity based on miRNA-135b expression, and the upstream modulator altered by arsenic stress could adversely influence the fish’s response to osmoregulation [13].

## 3. Temperature

Environmental temperature plays a major part in maintaining the life cycle of any fish. Due to global warming, it is important to understand the molecular mechanisms of miRNAs in adapting to the environment at different temperatures in the current context (Figure 2). miRNAs play an indicator role in the selection of molecular markers of cold-tolerant fishes [2]. miRNAs act more as a thermos-adaptor regulator in teleost. miRNAs have also been reported to have indirect effects on gene regulation through regulation of transcription factors, except for direct regulation of targets [35,36]. The change in water temperature has a significant influence on the physiology and behavior of fish and has been indicated to be a non-biological dominant factor for teleost [17]. A further rise in sea temperature might affect the fish’s life history. A temperature increase of several degrees significantly altered the expression profiles of short and long term miRNAs in the embryonic and larval stages of development [20].

### 3.1. miRNA Expression in Response to Temperature Changes

miRNA-155 was also found in upregulation in the liver of common carp and indicated a role of temperature tolerance [37]. In Atlantic cod, the stress-related miRNA-155 was increased in the total serum and serum-derived extracellular vesicles in the 9 °C group, while the growth-related miRNA-206 was increased in the 4 °C group. In the embryo and larval of Atlantic cod, the transcriptional expressions of miRNA-430 were not different at 4 °C and 9.5 °C. In addition, other transcription factors involved in the miRNA-430 cluster were affected by temperature changes [20]. Using binding site database of transcription factor, the transcription factor GATA1 (Figure 2) was bound to the upstream of the miRNA-27c gene, which indicated that the epigenetic temperature can regulate miRNA-27c expression [20]. Temperature rise can produce a stress response when fish experience higher temperature (9.5 °C), which leads to long-term changes in miRNA-449a expression [20].

Extracellular vesicles were used as a new biomarker to assess the health of fish in environmental feeding temperatures. The temperature-mediated effects were observed in miRNA-206 expression, which may indicate better growth conditions when raised at a lower temperature of 4 °C [38]. In the larvae of Senegalese sole (*Solea senegalensis*), ability in stimulating myogenesis could explain the differences in the transcription levels of miRNA-199, miRNA-206 and miRNA-17 between the two temperature groups at hatching. The miRNA levels were higher in the 21 °C than in the 15 °C group, which was consistent with the observed muscular phenotype [18]. Myoblast determination protein (MyoD) (Figure 2) was found to regulate the expression of miRNA-133 and miRNA-206. When the embryos of sole were incubated at 15 °C and 21 °C, the expression levels of miRNA-133 were higher at 21°C than at 15°C at the 20S stage of MyoD compared with the expression levels of miRNA-206 [18]. During incubation, higher expression levels of miRNA-17a, miRNA-181 and miRNA-206 in the 21 °C group may promote myogenic program. The growth activity of higher embryonic temperature was enhanced compared with that at 15 °C [18]. miRNA abundances associated with lipid metabolism vary in different temperature groups. The lead chain expressions of liver-specific miRNA-122 were higher at 15 °C than that at 21 °C. Considering the role of miRNA-122 in fatty acid oxidation and energy production, temperature influences lipid metabolism and energy utilization and affects the growth of larvae [18].

### 3.2. miRNA Functions and Its Target mRNAs under Different Temperature Environment

In zebrafish, miRNA-29 was involved in acute environmental stress (mostly cold stress) and targeted a core clock gene, period circadian regulator 2 (PER2) (Figure 2), to increase cold tolerance of the fish larvae [17]. In the liver of genetically improved farmed tilapia, miRNA-1338 could upregulate the expressions of growth hormone inducible transmembrane protein (GHITM) (Figure 2) in response to heat stress, suggesting the involvement of miRNA-1338 in regulation of liver growth and temperature control [16]. The upregulation of miRNA-99 under heat stress was related to the adaptive regulation of target gene heme oxygenase 1 (HMOX1) (Figure 1).

In our previous research on the liver of obscure pufferfish (*Takifugu fasciatus*) [39], comparative quantitative proteomic analysis focused on the liver in response to hypothermia using isobaric tags for a relative and absolute quantitation (iTRAQ) sequencing technique. HSP90, glutathione s-transferase (GST), ras-related protein (RAP1A), d-amino acid oxidase (DAAO) and cytochrome oxidase subunit 5a (COX5A) were identified in upregulation and ribosomal protein s6 kinase 2 alpha (RPS6KA), filamin B (FLNB), erb-b2 receptor tyrosine kinase 2 (ERBB2), calcium-binding protein 39 (CAB39) and alpha-2-macroglobulin-like 1 (A2ML1) were identified in downregulation (Figure 2). miRNAs and their targets, miRNA-27a and HSP90; miRNA-133b and GST, miRNA-101-5p and RAP1A, miRNA-338 and COX5A, miRNA-106b and FLNB, miRNA-146b and A2ML1, were identified in the continuous sequencing results of mRNA-seq and miRNA-seq. Many differentially rich proteins (DAPs) have been identified that are involved in oxidative-stress, mitochondrial-enzymes, signal-transduction and other biological process. The potential relationship between miRNAs and mRNAs provided new information on teleost in adaptation to low temperature.

## 4. Oxygen Concentration

Oxygen serves as the energy resource in fish. In order to live in water, organisms generally need to adapt different survival strategies such as reducing metabolic rate and producing enough energy [40]. Severe hypoxia can lead to many more anaerobic metabolic pathways. Hypoxia can increase the utilization of the glycolytic pathway and reduce the use of aerobic pathways [41,42], miRNAs are involved in the aerobic and anaerobic contributions to total metabolism.

### 4.1. miRNA Expression in Response to Hypoxia

Hypoxia can lead to upregulation of miRNA-9 in pulmonary artery smooth muscle cells, which contradicts the results of studies on inhibition of miRNA-9 in brain tissues of hypoxic fish [43]. Let-7a, miRNA-122, and miRNA-9 were downregulated in the livers and brains of a hypoxic female medaka, while miRNA-2184 was upregulated in the testis of a hypoxic male medaka [43]. HIF-1a can alter the miRNA profiles of cells and affect the cell cycle progression, while hypoxia can block DNA replication in HIF-1a-mediated manners [24]. HIF-1a (Figure 3) can contribute to the upregulation of oxygen-induced miRNA-462/miRNA-731 clusters in zebrafish embryos. miRNA-462/miRNA-731 were upregulated in a HIF-1a-mediated manner and specifically targeted at dead-box helicase 5 (DDX5) (Figure 3) and protein phosphatase 1d magnesium-dependent (PPM1DA) under hypoxia conditions [24]. Endogenous miRNA-204 levels can change the expression of VEGF and affect the tilapia’s hypoxia tolerance performance by detecting hematological parameters and enzyme activity.

### 4.2. miRNA Functions and Its Target mRNAs under Different Oxygen Environment

In the ovarian follicular cells of marine medaka, miRNA-210 is one of the most extensive hypoxia responsive miRNAs. miRNA-210 induced hypoxia can lead to anti-apoptosis. The response of miRNA-210 was investigated when hypoxia disrupts the reproductive function [44]. Inhibition of miRNA-210 promoted follicular cell apoptosis and expressions of apoptosis marker (CASPASE3) in a hypoxic group, suggesting the regulatory effects of miRNA-210 on ovarian cell apoptosis of fish [44] (Figure 3). In addition, miRNA-9 and miRNA-181b induced by hypoxia play an important role in the apoptosis and steroidogenesis of fish ovaries.

miR-204-5p regulates cell apoptosis-related genes such as homeodomain-interacting protein kinase (HIPK) [22]. miRNA-351 influences the expression of vascular endothelial growth factor (VEGF) (Figure 3) under hypoxia conditions, which is responsible for the formation of physiological blood vessels and pathological angiogenesis [4]. miRNA-204 also influences antioxidant activity [4]. miRNA-204 knockdown can inhibit the elevation of hematological parameters induced by hypoxia stress, which means that miRNA-204 controls the oxygen carrying capacity of blood. Hypoxia can lead to a significant increase in activity levels of superoxide dismutase (*SOD*) (Figure 3). In marine Medaka, miRNA-21 and miRNA-29 are involved in the regulation of apoptosis and are found to be expressed in brain, liver and gonads.

In the livers of the Blunt snout bream (*Megalobrama amblycephala*), both miRNA-143 and miRNA-101 can directly target the key glycolytic enzyme hexokinase under histopathological hypoxia. Hexokinase is the initially important regulatory enzyme in the glycolytic pathway [45]. Hexokinase can accelerate the glycolysis rate of Atlantic cod under hypoxia stress and produce ATP in a hypoxic manner, which is consistent with the results of the study on blunt bream [46]. In zebrafish larvae, miRNA-462/miRNA-731 clusters are induced under hypoxia stress by the hypoxia induction factor 1a (HIF-1a) and play a role in cellular adaptations. β-carotene-9′, 10′-dioxygenase (BCO2) is the target of miRNA-125, and has been reported as a regulatory protein for oxidative stress during zebrafish development [23]. In the liver, spleen, muscle, brain, gill and heart of blunt snout bream, miRNA-462/miRNA-731 clusters play a role in regulating hypoxia response and are key signal mediators for hypoxic-mediated cellular adaptation [47]. The miRNA clusters are hypoxia responsive and severe hypoxia may lead to the transfer of aerobic and anaerobic contributions to total metabolism to escape the HIF-1-dependent pathway [24].

In the muscle of Largemouth bass (*Micropterus Salmoides*), miRNA-124 was detected by bioinformatics analysis and dual luciferase reporter assays as regulating lactate transportation on glycolysis under hypoxia by targeting monocarboxylate transporters 1 (MCT1) [21].

In the testicular tissues of medaka, euchromatic histone-lysine n-methyltransferase 2 (EHMT2), miRNA-125 is an epigenetic regulator of transgenerational reproductive damage caused by hypoxia [22].

At the larval level of Tibetan naked carp, *Gymnocypris przewalskii*, miRNA-145 and miRNA-125 were distinctly expressed in the thermal stress group and control group, and played an important role in the cellular response to stress (heat shock and oxidative stress) [48].

In our research group, in the livers of Dark barbel catfish *(Pelteobagrus vachelli)* [42], the sequencing technique was used to characterize mRNA-seq and miRNA-seq to elucidate the molecular mechanisms of hypoxia adaptation. Compared with control fish (oxygen concentration, 0.7 mg/L), three miRNAs (miRNA-210-5p, miRNA-143 and miRNA-27b) were significantly identified to upregulate and eight miRNAs (miRNA-17-5p, miRNA-301c-3p, miRNA-16a-3p, miRNA-20a, miRNA-338, miRNA-3618, miRNA-338-3p and miRNA-301a) were significantly identified to downregulate in the groups of experimental fish (oxygen concentration, 6.8 mg/L). In our results, 162 miRNA-mRNA key pairs with negative correlation were identified with the involvement of 18 different expressions of miRNAs and 107 different expressions of mRNAs in total. Among them, metabolism related genes: 6-phosphofructokinase (PFKL), hexokinase (HK), lactate dehydrogenase (LDH), phosphoglycerate mutase (PGAM), lipoprotein lipase (LPL); Cancer related genes: vascular endothelial growth factor (VEGF), erythropoietin (EPO), apoptosis regulator BCL-2 (BCL2), von Hippel-Lindau disease tumor suppressor (VHL), transferrin receptor (TFRC), solute carrier family member 1 (SLC2A1), death-associated protein kinase (DAPK), transcription factor AP-1 (JUN), angiopoietin-like 4 (ANGPTL4), carbonic anhydrase (CA); Signal transduction related genes: activating transcription factor 2 (ATF2), cAMP response element modulator (CREM); insulin receptor substrate (IRS) related genes: dual specificity phosphatase (DUSP8), serine/threonine kinase (Akt), 5′-AMP-activated protein kinase, regulatory gamma subunit (PRKAG2), were found as potential mRNA targets in order to provide other layers of hypoxic response pathways (Figure 3). miRNA-mRNA pairs are used for bioinformatics analysis and miRNA prediction algorithms. The molecular mechanism of hypoxia adaptation in fish can be further understood through the comprehensive analysis of mRNA-seq and miRNA-seq in fish (Figure 3).

## 5. Feed

Live feeds for aquaculture (rotifers and Artemia) are nutritionally inferior to reference feeds (copepods). Specific enrichment or modification or supplementation of live feeds can alter miRNA levels in larvae which may provide a strategy for improving larval aquaculture [30]. miRNAs also have significant impacts on muscle plasticity and recovery after dietary restriction. Short periods of food restriction before feeding mainly affected changes in rapid muscle, muscle fiber diameter and miRNA expressions [31].

### 5.1. miRNA Expression in Response to Feeding

In skeletal muscle of Chinese perch, *Siniperca chuatsi*, refeeding induced the coordinated expression of several miRNAs (miRNA-10C, miRNA-107a, miRNA-133A-3P, miRNA-140-3P, miRNA-181A-5P, miRNA-206 and miRNA-214), which were involved in the strong recovery of myogenesis during feeding. In fast skeletal muscle of grass carp (*Ctenopharyngodon idella*), based on analysis of the rapid refeeding response after fasting, eight miRNAs (miRNA-1a, miRNA-181a, miRNA-133a, miRNA-214, miRNA-133b, miRNA-206, miRNA-146, and miRNA-26a) were involved in rapid skeletal muscle contractile growth. Recorded changes in miRNA expression levels have been shown to be associated with strong recovery of myogenesis [32]. In the muscle of rainbow trout, after eight weeks of short-term methionine restriction (MR) feeding, miRNA-133a abundance decreased and glucose tolerance overall increased. The observed reduction in miRNA-133a response to MR suggested that miRNA-133a may play a role during the overall growth reduction associated with methionine deficiency. This may be due to the reduced miRNA-133a signaling pathway at the muscle level, resulting in reduced muscle lipid accumulation and increased insulin sensitivity [49].

In the liver of juvenile rainbow trout, the expression of miRNA-122 was regulated by insulin changes from dietary macronutrient ratios, especially carbohydrate:protein and lipid:protein ratios. Transient elevations of miRNA-33 and miRNA-122b occurred four hours after feeding. Liver activity of the insulin signaling pathway and the expressions of lipogenic genes, including sterol regulatory element-binding transcription factor 1 (SREBF1), fatty acid synthase (FAS) and ATP-citrate lyase (ACLY) were both increased four hours after meal, while the expressions of lipid genes, carnitine palmitoyl-transferase 1a (CPT1a) and carnitine palmitoyl-transferase 1b (CPT1b), were significantly decreased (Figure 4). This suggested that the fat-generating effects of miRNA-33 and miRNA-122b may be conserved [50].

In the liver of rainbow trout, miRNA expression could be affected by selective breeding for better adaptation of plant-based feeds and total dietary substitution of plant components for fish meal and fish oil. Plant-based diet was found to have a significant induction effect on miRNA-33a and selective breeding had an effect on miRNA-122 and miRNA-128 [51]. Earthworm and duckweed are easy to be fed on by carp in natural water and were used as animal and plant feed to investigate the influence of feed type on antioxidant capacity of omnivorous fish. In the liver of common carp, the expression of miRNA-137 was downregulated when carps fed on duckweed and earthworms, suggesting that a mixed diet could reduce oxidative stress in common carp. Omnivorous carp exhibit greater antioxidant capacity when feeding on a mix of animals and plants. Five miRNAs including miRNA-137, miRNA-143, miRNA-146a, miRNA-21 and miRNA-125b have been reported to be associated with oxidative stress [52].

### 5.2. miRNA Functions and Its Target mRNAs under Feeding

The miRNAs could be involved in regulating the growth of fish muscle cutters. miRNA-10c was involved in response to nutritional restrictions and refeeding. The expression of miRNA-10c was significantly increased in one hour after fast muscle refeeding, which may be related to the use of amino acids and lipids. Upregulation of miRNA-10c can regulate the target gene Diacylglycerol O-Acyltransferase 2 (DGAT2) and catalyze the last step of triglyceride synthesis [53]. In both fast and slow muscle of Juvenile pacu (*Piaractus mesopotamicus*), a short period of food restriction showed that the expression of miRNA-1, miRNA-206, miRNA-199 and miRNA-23a significantly increased in fast muscle and the expression of miRNA-1 and miRNA-206 significantly decreased in slow muscle, and their targets (IGF-1 for miRNA-1, miRNA-206 and miRNA-199; mTOR for miRNA-199; and MFbx and PGC1a for miRNA-23a) presented negatively correlated expression profiles.

In the liver of blunt snout bream, upregulation of miRNA-30c and downregulation of miRNA-145 and miRNA-15a-5p were identified as responding to high fat diet throughput sequencing analysis. Six putative lipid metabolism-related target genes including Fetuin-B (FETUB), cytochrome P450 7A1 (CYP7A1), dehydrogenase 1 beta subcomplex subunit 2 (NADH), 3-oxoacid CoA transferase 1b (OXCT1), stearoyl-CoA desaturase (SCD) and fatty-acid synthase (*FAS*) were identified as potentially important roles in the development of diet-induced liver steatosis [54].

In Atlantic cod larvae, when the fed with reference feeds (zooplankton) and aquaculture feeds (rotifers and later brine shrimp), the first feed affected the expression of miRNA and its target. During the entire feeding period, miRNAs and their targets were identified and differentially expressed between the two feeding groups: miRNA-9 with CELA2A (chymotrypsin-like elastase family, member 2A), IGFALs (insulin-like growth factor binding protein), RGR (retinal G protein-coupled receptor), PHKG1 (phosphorylase kinase, gamma 1); miRNA-19a with col1a2 (collagen alpha-2 chain); miRNA-130b with SLC40A140a1 (solute carrier family 40 members 1, Fe-regulated transporter); miRNA-146 with mknk1 (MAP kinase interacting serine/threonine kinase 1); miRNA-181a with DUSP5 (dual-specificity phosphatase 5), CALML4 (calmodulin-like protein 4); miR-206 with Myb (transcriptional factor myb) [30].

miRNAs also play roles in the metabolism of other tissues or cells. miR-29a and miR-223 were involved in protein and glucose metabolism during the early stages of refeeding of Arctic Charr [55]. miR-101b mediated lipid deposition and metabolism in the primary hepatocytes of yellow catfish *Pelteobagrus fulvidraco* [56].

Starvation can induce autophagy and disrupt antioxidant systems. Autophagy related genes were activated and starvation disrupted intestinal homeostasis in the Chinese perch [57]. miR-252 increased autophagy and antioxidant enzyme activity by targeting PI_3_K to enhance the tolerance of *Penaeus vannamei* to ammonia nitrogen stress [58], while few miRNA studies have reported on autophagy when fish starved. Future miRNA studies can focus on this area.

## 6. pH

In the kidney of Nile tilapia, in vitro and in vivo, miRNA-21 substantially participates in and regulates alkaline stress by targeting 3′- untranslated regions (UTRs) of vascular endothelial growth factor (VEGF) mRNAs. Under alkaline stress, miRNA-21 participates in a regulatory loop that causes rapid changes in gene expression. Alkalinity stress resulted in a significant decrease in miRNA-21 level. The loss of miRNA-21 function can reduce the production of renal ROS in tilapia and increase the activities of catalase (CAT), glutathione peroxidase (GSH-Px) and superoxide dismutase (SOD). A direct association between miRNA expression and alkali resistance was revealed in tilapia. miRNA-21 will be developed as a maker for marker-assisted selection of alkaline-resistant fish.

The main physiological effects of aluminum-rich acidic water on fish are disorders in breathing gas transfer and ion regulation. In the muscle cells in wild Atlantic salmon exposed to acidic aluminum-rich water, the significantly differential miRNAs regarded as toxicological stressors including four miRNA (miRNA-122, miRNA-217, miRNA-133b, miRNA-216b) downregulated and fourteen miRNA (miRNA-204, miRNA-365, miRNA-728, miRNA-499, miRNA-23a, miRNA-338, miRNA-192, miRNA-733, miRNA-202, miRNA-363, miRNA-135a, miRNA-145, miRNA-15a, let-7h) upregulated in the exposed group (an average pH = 5.60 in the acid Aluminum-enriched tank) compared with control water (pH = 7.87), which experienced changes in many physiological responses. miRNA-23a has an indicative role in response to environmental persistent flame retardants (perfluoro-octane sulfonate). miRNA-145 is involved in the paracrine mechanism of zebrafish. miRNA-202 has a regulatory effect on psychosocial stress [59].

## 7. Environmental Chemicals and Sea Water Metal Elements

Cadmium stress induced miRNA-122 by a novel regulator of metallo-thionin in the liver of genetically improved farmed tilapia, which may promote metallo-thionin expression levels by directly targeting the 3′-UTR of metallo-thionin mRNA. miRNA-122 silencing increased the level of metallo-thiocyanin mRNA in tilapia and regulated signals related to metabolic regulation and antioxidant defense systems. Moreover, the overexpression of miRNA-125 increased the anti-apoptotic protein B-cell lymphoma 2 (BCL2) (Figure 5) and reduced the pro-apoptotic protein BCL2; for example, some apoptotic regulatory proteins helped to alleviate cadmium-induced apoptosis [16]. The role of miR-451 in mediating cadmium induced head kidney injury in common carp was studied via the targeting of cacna1ab through autophagy pathways [60].

miRNA -155 and CYB561D2 were evaluated under stress of fipronil in zebrafish, and cytochrome b561 domain-containing protein 2 (CYB561D2) is the major target of miRNA-155. After stimulation of fipronil, the expression of miRNA-155 was downregulated while mRNA and protein levels of CYB561D2 were upregulated in a dose-dependent manner. Dual-luciferase report analysis showed that miRNA-155 interacts with CYB561D2 3’-UTR. In addition, miRNA-155 can be considered as a potential biomarker of fipronil toxicity (Figure 5) [27].

Triclosan (TCS) is widely used as a broad-spectrum antibacterial agent in personal care products and medical devices. TCS was frequently detected in soil and municipal sewage, resulting in high bioaccumulation in lakes and oceans [61]. TCS is widely used in personal care products, and its chronic exposure leads to severely toxic effects in fish. In the embryos of zebrafish exposed to a series of TCS concentrations (0, 62.5, 125 and 250 mg/L) from 6 to 120 h post fertilization (hpf), G-protein-coupled estrogen receptors (GPER) (Figure 5) were identified as transmembrane target molecules of TCS action and triggered the target pathway NRF2/MAPK/P53 via expression of miRNA-125 in the liver and brain. NRF2 is a key marker gene in the signaling pathway and TCS was a potential chemical regulatory factor of the NRF2 expression (Figure 5) [29].

The chronic toxicity of 1-methyl-3-octylimidazolium bromide (C8mimBr) was determined to further reveal the toxicological mechanisms of ionic liquids in the spleen of silver carp (*Hypophthalmichthys molitrix*). C8mimBr exposure reduced the expression of miRNA-125b and altered the expression of miRNA-143. Upregulation of miRNA-155 and miRNA-21 levels suggested that these miRNAs may be involved in the C8mimBr-induced inflammatory response. Chronic exposure to ionic liquid C8mimBr indicated that inducing inflammation and oxidative stress can be mediated by P38MAPK/NF-ΚB signaling pathways (Figure 5) [62].

Perfluoro-octane sulfonate (PFOS) concentrations in the liver of smallmouth bass (*Micropterus dolomieu*) and largemouth bass from New York State in the USA ranged from 9 to 315 ng/g wet weight, and the average concentrations of PFOS in the fish were 8850-fold greater than those in surface water [63]. miRNA expression changes during zebrafish development induced by PFOS. PFOS exposure induced significant changes of miRNA expression profiles. miRNA-19b-c, miRNA-181b, miRNA-19d and miRNA-735 showed significantly different expression patterns after PFOS exposure of 120 hours post-fertilization [64].

Iron is an essential metal cofactor. Normally enzymes are involved in many cellular functions, including energy production and cell proliferation. However, excessive iron concentrations can lead to an increase in oxidative stress and toxicity. In the brain of turquoise killifish (*Nothobranchius furzeri*), iron content (350 μg/g body weight iron dextran, 30 μg/g body weight deferoxamine and 50 μg/g body weight 4-OH-Tempol) can lead to an increased expression of miRNA-29; iron content can upregulate the expressions of miRNA-29 and in turn targets the 3’-UTR mRNA of iron-reactive element binding protein 2 (IREB2), thereby reducing the iron intake [28].

Environmental trace metals (Cu) can damage the normal function of fish in the olfactory system of zebrafish. Differentially expressed miRNAs were found in a dose-response manner corresponding to three increasing Cu concentrations (concentrations of 6.3, 16 and 40 ppb Cu). Many deregulated miRNAs (let-7, miRNA-7a, miRNA-128, and miRNA-138) are involved in neurogenesis, suggesting that copper mediates toxic effects by interfering with neurogenesis. In addition, some miRNAs including miRNA-203a, miRNA-199, miRNA-16a, miRNA-16C and Mir-25 may lead to decreased mRNA levels of their host genes through post-transcriptional mechanisms, which are involved in olfactory signal transduction pathways. Environment-related copper concentrations altered the expression of different miRNA amounts, and the predicted gene targets were preferentially involved in the key neural processes. The use of miRNAs can be extended as biomarkers for environmental applications involving metal exposure [65].

In our previous work, in the liver of *T. fasciatus* [66], the juvenile fish were exposed to control, 20 or 100 μg copper nanoparticles (Cu NPs)/L for thirty days. With an increase in Cu NPs dose, the activities of succinate dehydrogenase (SDH) (Figure 5), NKA and cytochrome c (CYT-C) decreased in mitochondria, which was accompanied by the increasing concentration of CYT-C in cytosol (Figure 5). The expressions of apoptosis-related genes (P53, CASPASE-3, CASPASE-9 AND BAX) were increased and BCL2 expression decreased with the increase of Cu NPs dose. The physiological indicators of immune-response related genes were indicated include heat shock protein 70 (HSP70), heat shock protein 90 (HSP90), immunoglobulin M (IgM) and lysozyme (LZM). The mRNA expressions of HSP70, HSP90, IgM and LZM were increased after Cu NPs exposure. The associated clusters among miRNA15-HSP70, miRNA27-HSP90, miRNA150-IgM, miRNA143-CASPASE3 and miRNA96-CASPASE9 were identified when exposed to an increase in Cu NPs dose.

## 8. Conclusions

miRNAs control several key cellular processes and have emerged as a fundamental epigenetic regulatory mechanism. The understanding of the role of these small molecules is essential in environmental stress mechanisms which have expanded significantly over the past few years. In teleost, miRNAs have aberrant processing and expression profiles in response to different environmental stress patterns; in addition, circulating miRNA profiles are also affected, making them potential biomarkers with possible diagnostic applications. Therefore, miRNA-based targeting markers offer promising prospects. Further studies are needed to define specific processes and molecular interactions in order to use this knowledge to improve the management of fish in response to environmental stress.

## 9. Perspective

Other environmental factors, such as photoperiod, have long-term effects on the gene expression of miRNA. Previous studies of miRNAs have focused on the influences of photoperiod on plants [67,68]. Photoperiod has important effects on fish development, but these effects of miRNAs have rarely been described.

To understand the regulatory role of miRNA in the environment, the best approach is to identify its target. Identification of miRNA targets by experimental methods may be the next important step in studying the role of miRNAs. The silico prediction of the target gene was refined by using the gene expression data obtained in parallel in the same sample, and only a few studies have carried out experimental verification of the target gene. Methods were used to validate target genes in examining the potential negative regulation of a particular miRNA toward a particular putative target transcript. A combination of multiple approaches (single-target validation, genome-wide approaches, and cross-species comparisons) can help identify the target transcripts of environmental-reactive-related miRNAs in teleost [69].

miRNAs have different functions in different fish species, so the function of miRNAs should be further explored, compared with that of mammals. Differences in miRNA expression varied with fish species when compared with chicken or mouse, and these data fit well with the expectation that miRNA expressions may be linked with differences in animal physiology. Interspecies changes in gene regulation have been identified as the driving force behind structural changes in the body, while transcriptional regulation has recently been identified as a mechanism involved in evolution [70].

This is a tilapia miRNA database (miRTil) and more databases need to emerge which will predict the target genes of environmental impacts. miRTil is a complementary data source for miRbase, in order to integrate information on miRNAs in Nile tilapia and to assist scientists who are interested in understanding the regulatory mechanisms of miRNAs. The database specializes in providing miRNA information obtained from RNA-seq technology and the results of bioinformatics analysis, such as target prediction and gene expression profiling. This data will allow expression of miRNA genome segments preserved in vertebrate species, as well as miRNA-centric views of the evolutionary forces of Nile tilapia lineages and other teleost [71]. Gene editing can be used for miRNA research. Many miRNA-mediated biological characteristics can be used to study fish in different farmed environments. Genome editing is the best way to increase or decrease miRNA expression and can contribute to improving important characteristics that can enhance global aquaculture production.

Very little genomic data is available with well-defined characteristic genes for fish. The challenge ahead is to use next-generation sequencing and computing to understand the large number of non-model fish genomes, their overall gene functions and their evolutionary patterns [72]. miRNAs can silence the specific gene expression in complementary mRNAs from translational repression and decay [73]. Splicing factor proline protein can regulate miRNA silencing on specific binding sites and globally impact miRNA-dependent gene expression programs [74]. Many studies have also reported that miRNAs can provide reversible gene silencing mechanisms in aestivation and hibernation and are involved in cellular adaptation to specific demands under stressful conditions [75,76]. In recent years, epigenetic factors have been thought to overwhelm gene expression and corresponding protein products. For example, epigenetic and energetically costly transcription or translation repression support metabolic suppression in chronically hypoxic goldfish. Goldfish can rely on transcriptional silencing of chronically hypoxic brain and heart via hypermethylation after being transiently hypomethylated, which suggested that transcriptional modifications support metabolic suppression, but require a long hypoxia exposure to occur in these critical tissues [77].miRNA is involved in post-transcriptional genes, but its mechanisms in disease conditions are still largely unclear. Each miRNA has several hundred predicted target miRNAs, but only a small number of interactions have been experimentally validated (Figure 6).

Very few genetic studies have reported that environmental stress can influence any miRNA mediated silencing machinery in fish, demonstrating a major gap in the scientific literature in this field compared with mammal biology. Dicer endoribonucleases generate small RNAs for miRNA and RNA interference pathways [78]. Mammalian dicer supported the essential gene-regulating miRNA pathway, but how it is specifically adapted to miRNA biogenesis is unknown [79]. Mature miRNA in the cytoplasm are known to be associated with Argonaute proteins (AGOs) in the RNA induced silencing complex (RISC); the RISC forms a complex with the miRNA by guiding the effector complex to the target mRNA for gene regulation [80]. Recent progress in dicer-related research, mature miRNA enriched in the nucleus associated with RISC proteins, and potential RNA interference pathways will be discussed in aquatic species.

## Figures and Tables

**Figure 1 biology-12-00388-f001:**
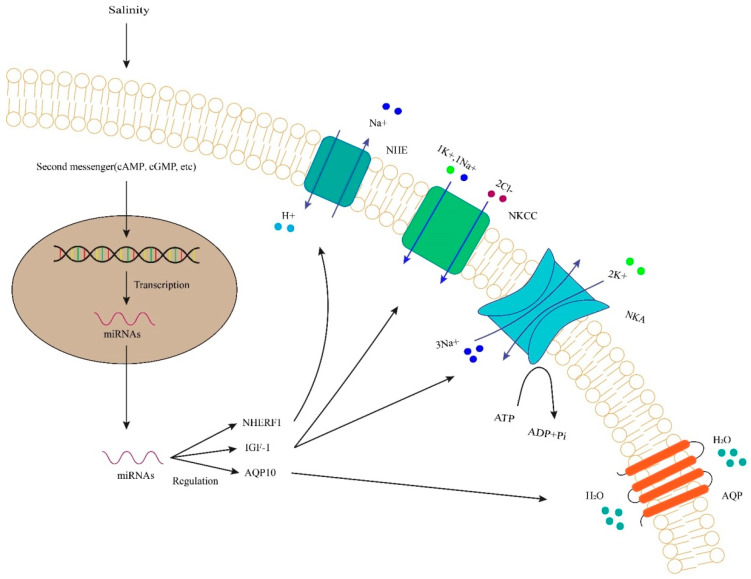
Regulation of NHE, NHERF1, NKA, NKCC, IGF-1 and AQP, etc., by miRNAs under salinity in fish. Second messengers induce the transcription of DNA. miRNAs regulate gene expression and function in target genes. They regulate transcription of the target genes and protein synthesis. NHE, mean Na^+^/H^+^-exchanger; NHERF, Na^+^/H^+^ exchanger regulatory factor 1; NKCC, mean Na^+^-K^+^-2Cl- cotransporter; NKA, mean Na^+^/K^+^-ATPase; AQP, mean Aquaporin; IGF, mean insulin growth factor.

**Figure 2 biology-12-00388-f002:**
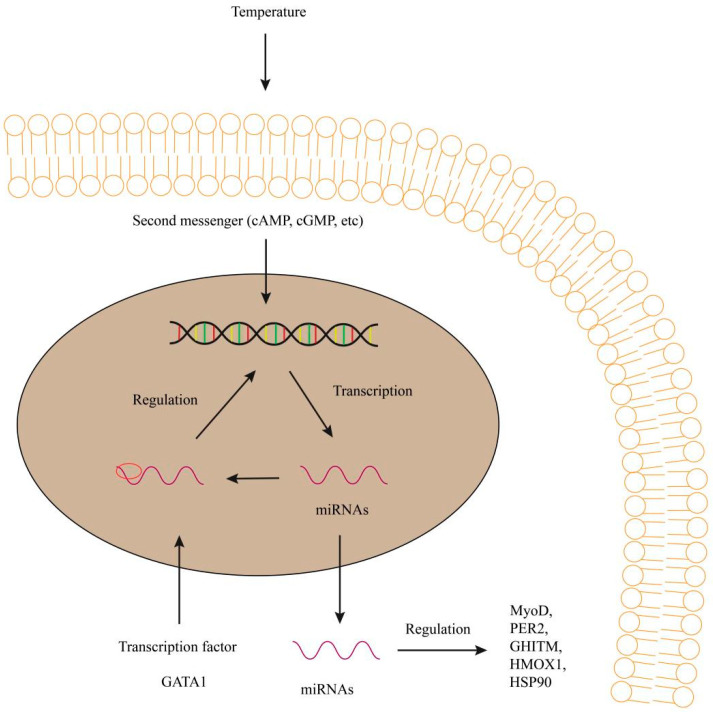
Regulation of GATA1, MyoD, PER2, GHITM, HMOX1 and HSP90 etc. by miRNAs under temperature in fish. miRNAs can negatively regulate the transcription of DNA induced by second messengers. miRNAs regulate gene expression and function in target genes. They regulate transcription of the target genes and protein synthesis. GATA1, mean GATA binding protein 1; MyoD, Myoblast determination protein; PER2, mean period circadian regulator 2; GHITM, mean growth hormone inducible transmembrane protein; HMOX1, mean heme oxygenase 1; HSP90, mean heat shock protein 90.

**Figure 3 biology-12-00388-f003:**
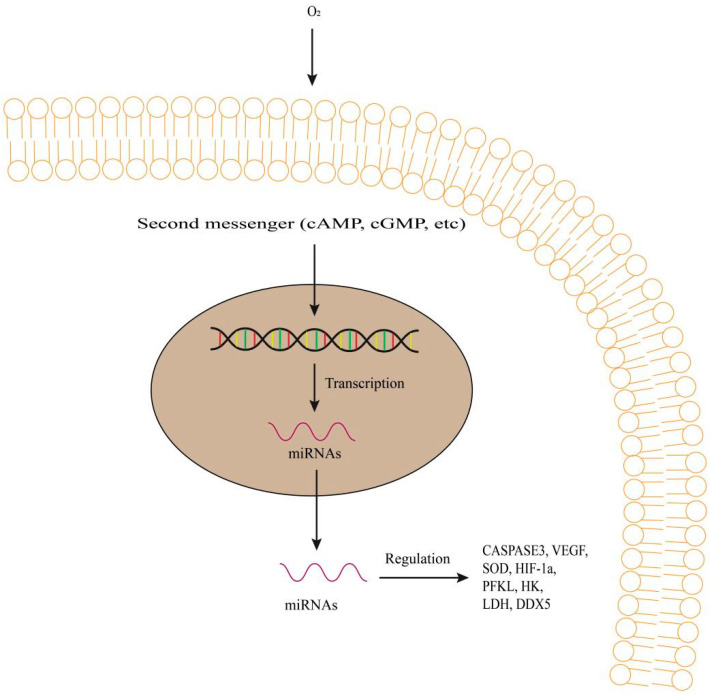
Regulation of CASPASE3, VEGF, SOD, HIF-1a, PFKL, HK, LDH and DDX5, etc., by miRNAs under oxygen concentration in fish. Second messengers induce the transcription of DNA. miRNAs regulate gene expression and function in target genes. They regulate transcription of the target genes and protein synthesis. VEGF, mean vascular endothelial growth factor; SOD, superoxidase dismutase; HIF-1a, mean hypoxia-inducible factor 1a; HK, mean hexokinase; LDH, mean lactate dehydrogenase; DDX5, mean DEAD-Box Helicase 5.

**Figure 4 biology-12-00388-f004:**
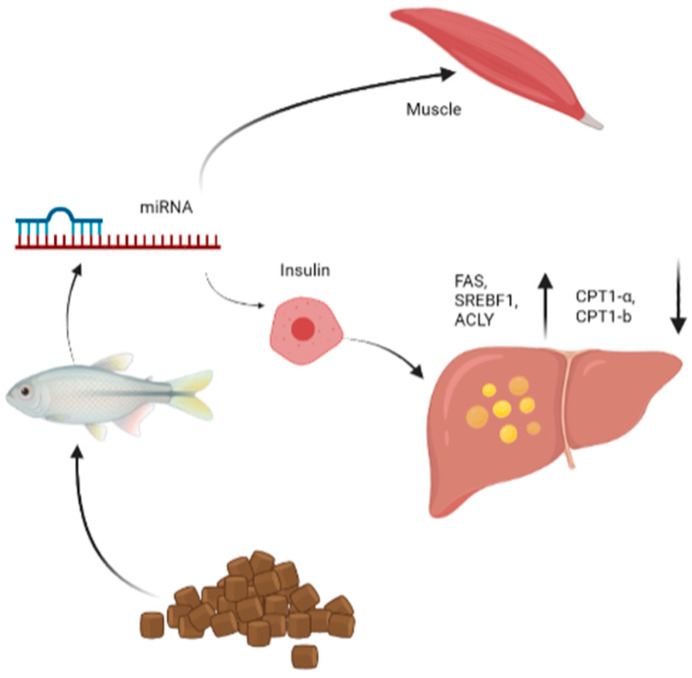
Role of microRNAs in the regulation of insulin and liver metabolism including the upregulation of FAS, SREBF1, ACLY and downregulation of CPT1a, CPT1b. FAS, SREBF1, ACLY, CPT1a and CPT1b, etc., are regulated by miRNAs under feed in fish. FAS, mean Fatty acid synthase; SREBF1, sterol regulatory element binding transcription factor 1; ACLY, mean ATP citrate lyase; CPT1a, mean carnitine palmitoyl-transferase 1a; CPT1b, mean carnitine palmitoyl-transferase 1B.

**Figure 5 biology-12-00388-f005:**
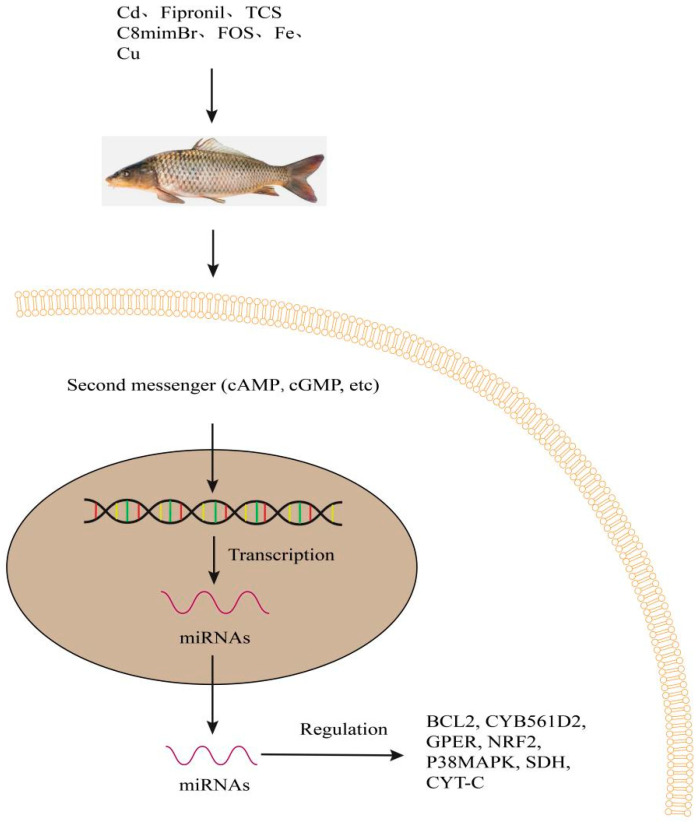
Regulation of BCL2, CYB561D2, GPER, NRF2, P38MAPK, SDH and CYT-C, etc., by miRNAs under environmental chemicals and sea water metal elements in fish. Second messengers induce the transcription of DNA. miRNAs regulate gene expression and function in target genes. They regulate transcription of the target genes and protein synthesis. BCL2, mean B-cell lymphoma 2; CYB561D2, Cytochrome B561 Family Member D2; GPER, mean G protein-coupled estrogen receptor; NRF2, mean nuclear factor erythroid-2-related factor 2; P38MAPK, mean p38 mitogen-activated protein kinases; SDH, mean succinate dehydrogenase; CYT-C, mean cytochrome C; cAMP, cyclic adenosine monophosphate; cGMP, cyclic guanosine monophosphate.

**Figure 6 biology-12-00388-f006:**
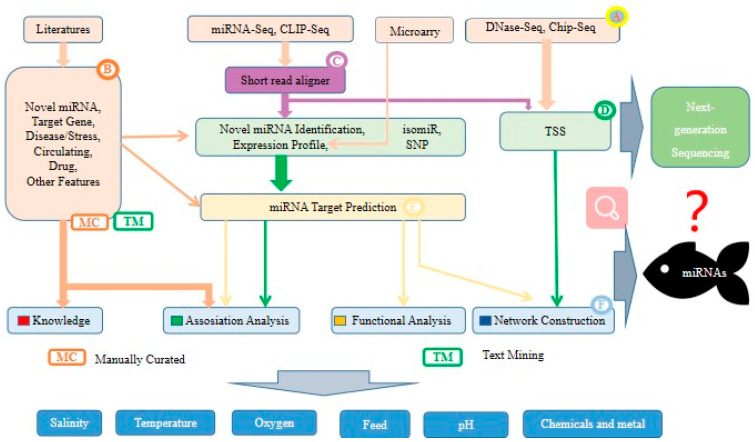
Standard analysis process and prospect of environment-related miRNAs in fish. (A) Data sets download, (B) search of background knowledge, (C) read alignment, (D) identification and characterization of known and novel miRNAs, (E) target prediction, (F) downstream analysis.

**Table 1 biology-12-00388-t001:** Browsed miRNA information from the teleost miRBase including species, ID abbreviation, number of hairpin precursor entries, mature miRNA sequences, mature entries, dead entries and miRNA entries (http://www.mirbase.org/index.shtml, accessed on 7 June 2018). 17 January 2023.

Species	ID Abbreviation	Number of Hairpin Precursor Entries	Number of Mature miRNA Sequences	Number of Mature Entries	Number of Dead Entries	Number of miRNA Entries
Zebrafish (*Danio rerio*)	dre-mir dre-let	398	375	141	102	1593
Nile tilapia (*Oreochromis niloticus*)	oni-mir	812	695	0	0	1507
Atlantic cod (*Gadus morhua*)	gmo-mir	401	516	0	0	917
Atlantic salmon (*Salmo salar*)	ssa-mir ssa-let	371	498	0	0	869
Burton’s mouth-brooder (*Astatotilapia burtoni*)	abu-mir abu-let	298	236	0	0	534
Channel catfish (*Ictalurus punctatus*)	ipu-mir ipu-let	281	205	0	0	486
Nyassa blue cichlid (*Metriaclima zebra*)	mze-mir	256	184	0	0	440
Fairy cichlid (*Neolamprologus brichardi*)	nbr-mir nbr-mir	251	182	0	0	433
Victoria cichlid (*Pundamilia nyererei*)	pny-mir	250	182	0	0	432
Japanese rice fish (*Oryzias latipes*)	ola-mir ola-let	168	146	0	0	314
Common carp (*Cyprinus carpio*)	ccr-mir ccr-let	134	146	0	0	280
Pufferfish (*Fugu rubripes*)	fru-mir fru-let	133	108	6	3	255
Spotted green pufferfish (*Tetraodon nigroviridis*)	tni-mir tni-let	132	109	2	0	245
Atlantic halibut (*Hippoglossus hippoglossus*)	hhi-mir hhi-let	40	37	0	1	78
Olive flounder (*Paralichthys olivaceus*)	pol-mir pol-let	20	38	0	0	58
Electric eel (*Electrophorus electricus*)	eel-mir	20	34	0	0	54

**Table 2 biology-12-00388-t002:** Overview of teleost fish studies on selective miRNAs concerning environmental stress including author information, species, tissue, stress type, miRNA list, target genes and function. 17 January 2023.

Species	Tissue	miRNA	Target-Genes	Function	
Salinity					
Japanese eel *Anguilla japonica*	Gills	miRNA-200b-3p	slc17a5	Serve as organic osmolytes to regulate cellular osmolality and against hyperosmotic stress	[10]
Japanese eel	Gills	miRNA-29b-3p	Krueppel-like factor 4 (KLF4)	Increase chloride cell densities in fish gills and are responsible for ion transport	[10]
Zebrafish	Embryos	miRNA-8 miRNA-200	NHERF1	A regulator of apical trafficking of transmembrane ion transporters	[11]
Tilapia	Kidney	miRNA-30c	HSP70	Responds to osmotic stress	[12]
Atlantic killifish (*Fundulus heteroclitus*)	Kidney	miRNA-135b	NR3C2	Regulation during salinity acclimation	[13]
Atlantic killifish	Kidney	miRNA-135b	Potassium voltage-gated channel	Important mediators of hyperosmotic response	[13]
Nile tilapia	Gills	miRNA-429	OSTF1	Lead to changes in the ionic concentration and osmotic stress; Influence the regulation of plasma osmolality and ion concentration responding to osmotic stress	[14]
Tilapia	Skeletal muscle	miRNA-206	IGF-1	Upregulate the expression of transporters such as NKA and NKCC	[15]
Genetically improved farmed tilapia	Liver	miRNA-PC-5p-27517/3p-50929	AQP10a	Mediate AQP10a that absorb and excrete water	[16]
Temperature					
Zebrafish	Larvae	miRNA-29	Period Circadian Regulator 2 (PER2)	Enhance cold tolerance of the fish larvae	[17]
Senegalese sole	Embryos	miRNA-133 miRNA-206	Myoblast determination protein (MyoD)	Stimulate myogenesis	[18]
Emerald notothen (*Trematomus bernacchii*)	Gill	miRNA-21	Forkhead box protein (FOX)	Returning to a physiological state of thermal acclimation	[19]
Atlantic cod	Embryo and larval	miRNA-27c	GATA-binding factor 1	Epigenetic modulation by temperature	[20]
Genetically improved farmed tilapia	Liver	miRNA-1338-5p	Growth hormone inducible transmembrane protein (GHITM)	Regulate cell growth and oxidative stress	[16]
Genetically improved farmed tilapia	Liver	miRNA-99	Heme oxygenase 1 (HMOX1)	Regulate heat stress	[16]
Oxygen concentration					
Largemouth bass (*micropterus salmoides*)	Muscle	miRNA-124	Monocarboxylate transporters 1 (MCT1)	Regulate lactate transportation under hypoxia	[21]
Medaka (*Oryzias melastigma*)	Testicular tissues	miRNA-125-5p	Euchromatic histone-lysine N-methyltransferase 2 (EHMT2)	The epigenetic regulator of transgenerational reproductive impairment	[22]
Zebrafish	Larvae	miRNA-125	β-carotene 9,10-dioxygenase (BCO2)	Regulate oxidative stress	[23]
Medaka	Ovarian follicular cells	miRNA-351	Homeodomain-interacting protein kinase (HIPK)	Regulate cell apoptosis	[22]
Zebrafish	Larvae	miRNA-462 miRNA-731	Hypoxia-inducible factor 1a (HIF-1a), DEAD box protein 5 (DDX5), Protein phosphatase 1D (PPM1DA)	Regulate cellular adaptations	[24]
Nile tilapia	Kidney	miRNA-21	Vascular endothelial growth factor (VEGF)	Involve in regulating the alkalinity stress	[25]
Genetically improved farmed tilapia	Liver	miRNA-122	Metallothionein	Involved in modulating the cadmium-stress response	[26]
Environmental chemicals and sea water metal elements					
Zebrafish	Whole body	miRNA-155	Cytochrome b561 domain-containing protein 2 (CYB561D2)	A potential biomarker for fipronil toxicity	[27]
Turquoise killifish (*Nothobran chius furzeri*)	Brain	miRNA-29	Iron responsive element binding protein 2 (IREB2)	Reducing the toxicity of iron intake	[28]
Zebrafish	liver and brain	miRNA-125	G-protein-coupled estrogen receptors (GPER)	Trigger the target pathway NRF2/MAPK/P53 as a potential regulatory factor	[29]
Feed					
Atlantic cod (*Gadus morhua*)	Larvae	miRNA-9	Chymotrypsin-like elastase family, member 2A (CELA2A); insulin-like growth factor binding protein (IGFALS); retinal G protein-coupled receptor (RGR); phosphorylase kinase, gamma 1 (PHKG1)	Improve larviculture	[30]
Atlantic cod	Larvae	miRNA-19a	Collagen alpha-2 chain (COL1A2)	Improve larviculture	[30]
Atlantic cod	Larvae	miRNA-130b	Solute carrier family 40 member 1, Fe-regulated transporter (slc40a1)	Improve larviculture	[30]
Atlantic cod	Larvae	miRNA-146	MAP kinase interacting serine/threonine kinase 1 (MKNK1)	Improve larviculture	[30]
Atlantic cod	Larvae	miRNA-181a	Dual-specificity phosphatase 5 (DUSP5); Calmodulin-like protein 4 (CALML4)	Improve larviculture	[30]
Atlantic cod	Larvae	miRNA-206	Transcriptional factor myb (MYB)	Improve larviculture	[30]
Juvenile pacu (*Piaractus mesopotamicus*)	Muscle	miRNA-1 miRNA-206 miRNA-199	IGF-1	Affect fast muscle with changes in muscle fiber diameter	[31]
Juvenile pacu	Muscle	miRNA-199	mTOR	Affect fast muscle with changes in muscle fiber diameter	[31]
Juvenile pacu	Muscle	miRNA-23a	MFbx and PGC1a	Affect fast muscle with changes in muscle fiber diameter	[31]
Chinese perch (*Siniperca chuatsi*)	Muscle	miRNA-10c	Diacylglycerol O-acyltransferase 2 (DGAT2)	Regulate fish muscle growth	[32]

## Data Availability

Not applicable.

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
