# Peer review of "Profiling miRNAs of Teleost Fish in Responses to Environmental Stress: A Review"

_biology, 2023, doi:10.3390/biology12030388_

Round 1

Reviewer 1 Report

The authors present a review about profiling miRNAs of teleost fish as they respond to environmental stress. The topics seem very interesting; however, the presentation of the review requires work in order to clarify the objective and develop each section. In this reviewer's eyes, the manuscript is sloppy and the redaction lacks consistency.

My biggest concern is that there are paragraphs for which the citation does not match: for example, lines 195-197 (Qiang 2017a). In this reference, miRNA-21 and FOX were not validated.

In some instances, either the paragraph lacks a citation, or the closest one has no relation to the text, for example, lines 222-223; if the reference is Tse 2016, the information at this cite does not correspond to this section. The authors must review the entire manuscript.                                                                                 

Lines 20-22 should be deleted and the principal objective restructured

A preliminary paragraph is required to introduce each new section.

In the introduction, the authors should introduce the reader to the environmental factors, which are the apparent reason for the type of stress specified.

Fig 1A is not clear. Therefore, citing this image in the text is not relevant. Moreover, the authors should add a color code.

Check Typographic spaces in figures 2, 3, 4.

Please add figure legends to concisely explain each figure.

The authors should restructure Table 2, adding subtitles to classify the information, according to type of stress, as occurs in the text.

The authors transposed the information from the ovary to the testis, without defining the type of tissue (for example, lines 222-224).

Please add a conclusion section.

Author Response

Response to reviewer #1' comments

The authors present a review about profiling miRNAs of teleost fish as they respond to environmental stress. The topics seem very interesting; however, the presentation of the review requires work in order to clarify the objective and develop each section. In this reviewer's eyes, the manuscript is sloppy and the redaction lacks consistency.

Response: Many thanks for your kind comments. It's really a huge work about miRNA research of fish. We try our best to revise, summarize and develop each section in order to keep the consistency.

My biggest concern is that there are paragraphs for which the citation does not match: for example, lines 195-197 (Qiang 2017a). In this reference, miRNA-21 and FOX were not validated.

Response: Thanks for your reminding. In our resubmitted manuscript, the available references have been well revised and rechecked.

In some instances, either the paragraph lacks a citation, or the closest one has no relation to the text, for example, lines 222-223; if the reference is Tse 2016, the information at this cite does not correspond to this section. The authors must review the entire manuscript.

Response: We have checked all references in the resubmitted manuscript.

Lines 20-22 should be deleted and the principal objective restructured.

Response: Restructured.

A preliminary paragraph is required to introduce each new section.

Response: Thanks for your good comments. We reconstructed the corresponding parts and added the new sections especially in the topic about “Salinity” and “Temperature”.

In the introduction, the authors should introduce the reader to the environmental factors, which are the apparent reason for the type of stress specified.

Response:

Added in corresponding parts:

Environmental factors including salinity, oxygen concentration, temperature, feed, pH, environmental chemicals and sea water metal element may affect the transcriptional and posttranscriptional regulators of miRNAs contributing to nearly all biological processes. Environmental factors are the apparent reason for the type of stress specified. For fish, the survival of aquatic fish is constantly challenged by the changes in these environmental factors.

Fig 1A is not clear. Therefore, citing this image in the text is not relevant. Moreover, the authors should add a color code.

Response: Deleted.

Check Typographic spaces in figures 2, 3, 4.

Response: Corrected.

Please add figure legends to concisely explain each figure.

Response: Added and revised.

The authors should restructure Table 2, adding subtitles to classify the information, according to type of stress, as occurs in the text.

Response: Added.

The authors transposed the information from the ovary to the testis, without defining the type of tissue (for example, lines 222-224).

Response: Revised.

Please add a conclusion section.

Response: Added.

Reviewer 2 Report

Major comments

In this review, the authors summarized the profiling of numerous miRNAs of teleost fish in response to environmental stresses. On the other hand, I understand that it is inevitable that the manuscript is descriptive, citing original papers due to the nature of the “review article”, but even so, I feel that it just describes a part of each original paper. I recommend that you summarize a little more for each session (salinity, temperature, ..., etc.). Also, I could not read what the intention was for each paragraph within the sections to be summarized. Is it by fish species or miRNA groups? I feel that it would be easier for the reader to read if you describe the information in a consistent manner throughout the manuscript. Furthermore, I feel that the explanations in all of the figures are insufficient, resulting in unfriendly for readers. Additionally, I do feel that the relationship between sentences and figures is inconsistent. Based on the above, the manuscript needs to be substantially revised before it can be accepted for acceptance.

Minor comments

Introduction

L.30, twenty -> thirty?

L.46, Describe the date of access to the database.

Table 1, I do not know what each column means due to lack of explanation. Additionally, what is the difference between column 3 (Number of hairpin precursor entries) and column 4 (Number of hairpin precursor entry)?

L. 61-65, This sentence cites Fig1, but I didn't get the relationship between the sentence and Fig.1. Please rewrite it so that I can understand the intent of Fig1. 

Salinity

L. 137-139, I can't find any description of miRNA-206 being involved in growth, only stating that it is a target of IGF1 in the salinity section. 

Fig.2, I believe AQP has a six-transmembrane structure. I recommend that you draw a more accurate schematic diagram. Also, why didn't you put OSF1, NKA, and NKCC in Fig. 2? 

Temperature

L.166, miRNA-27c suddenly appeared in the manuscript, what is it? Please explain the simple background.

L.169, For miRNA-449a, same as above comment.

Oxygen concentration

L.224, sterogenesis -> steroidogenesis?

Food

L.372, miRNA-137 suddenly appeared in the manuscript, what is it? Please explain the simple background. 

pH

Figure 6, I feel the explanation is insufficient and uninformative. Also, I don't think the sentence (L.384-386) explains fig. 6.

L.402, Al-enriched, Al = Aluminum? Describe the abbreviation at first appearance.

Environmental chemicals and seawater metal element

L.431-432, What specific effects have been reported? Please cite if possible. 

Perspective

L.523, I don't feel that this sentence is appropriate as a statement describing Fig8.

Reference

Within Table 1, 

Yan et al, 2012 -> 2012a or 2012b?

Sakamoto et al., 2006, there is no information in the Reference list.

Qiang et al., 2017 -> 2017a?

Vasadia et al., 2019, there is no information in the Reference list.

Qiang et al., 2017 -> 2017b?

Qiang et al., 2017 -> 2017b?

Tse et al., 2016 -> 2015?

Qiang et al., 2017 -> 2017b?

Ripa et al., 2016 -> 2017?

Paula et al., 2016 -> 2017?

Paula et al., 2016 -> 2017?

Paula et al., 2016 -> 2017?

L.413-414, (Chen et al.,2016), there is no information in the reference list.

Please re-check your reference list carefully.

Author Response

Response to reviewer #2' comments

Major comments

In this review, the authors summarized the profiling of numerous miRNAs of teleost fish in response to environmental stresses. On the other hand, I understand that it is inevitable that the manuscript is descriptive, citing original papers due to the nature of the “review article”, but even so, I feel that it just describes a part of each original paper. I recommend that you summarize a little more for each session (salinity, temperature, ..., etc.). Also, I could not read what the intention was for each paragraph within the sections to be summarized. Is it by fish species or miRNA groups? I feel that it would be easier for the reader to read if you describe the information in a consistent manner throughout the manuscript. Furthermore, I feel that the explanations in all of the figures are insufficient, resulting in unfriendly for readers. Additionally, I do feel that the relationship between sentences and figures is inconsistent. Based on the above, the manuscript needs to be substantially revised before it can be accepted for acceptance.

Response: Thanks for your good comments. Some parts are really difficult to summarize and generalize. We try to and reconstructed and summarized in the conclusion and perspectives. The figure legends were added to concisely explain each figure.

Minor comments

Introduction

L.30, twenty -> thirty?

Response: Revised.

L.46, Describe the date of access to the database.

Response: Added.

Table 1, I do not know what each column means due to lack of explanation. Additionally, what is the difference between column 3 (Number of hairpin precursor entries) and column 4 (Number of hairpin precursor entry)?

Response: The parts of column 4 were deleted.

  1. 61-65, This sentence cites Fig1, but I didn't get the relationship between the sentence and Fig.1. Please rewrite it so that I can understand the intent of Fig1.

Response: Fig.1 was deleted.

Salinity

  1. 137-139, I can't find any description of miRNA-206 being involved in growth, only stating that it is a target of IGF1 in the salinity section.

Response: Deleted.

Fig.2, I believe AQP has a six-transmembrane structure. I recommend that you draw a more accurate schematic diagram. Also, why didn't you put OSF1, NKA, and NKCC in Fig. 2?

Response: Added.

Temperature

L.166, miRNA-27c suddenly appeared in the manuscript, what is it? Please explain the simple background.

Response: We addressed the two topics of expression of miRNAs in response to the environmental changes; miRNA functions and its target mRNAs under the specific environment. It's a huge job to introduce miRNA's background one by one, so we focused to write their associations.

L.169, For miRNA-449a, same as above comment.

Response: Above.

Oxygen concentration

L.224, sterogenesis -> steroidogenesis?

Response: Revised.

Food

L.372, miRNA-137 suddenly appeared in the manuscript, what is it? Please explain the simple background.

Response: Above.

pH

Figure 6, I feel the explanation is insufficient and uninformative. Also, I don't think the sentence (L.384-386) explains fig. 6.

Response: Added.

L.402, Al-enriched, Al = Aluminum? Describe the abbreviation at first appearance.

Response: Revised.

Environmental chemicals and seawater metal element

L.431-432, What specific effects have been reported? Please cite if possible. 

Response: Revised.

Perspective

L.523, I don't feel that this sentence is appropriate as a statement describing Fig8.

Response: Revised.

Reference

Within Table 1, 

Yan et al, 2012 -> 2012a or 2012b?

Sakamoto et al., 2006, there is no information in the Reference list.

Qiang et al., 2017 -> 2017a?

Vasadia et al., 2019, there is no information in the Reference list.

Qiang et al., 2017 -> 2017b?

Qiang et al., 2017 -> 2017b?

Tse et al., 2016 -> 2015?

Qiang et al., 2017 -> 2017b?

Ripa et al., 2016 -> 2017?

Paula et al., 2016 -> 2017?

Paula et al., 2016 -> 2017?

Paula et al., 2016 -> 2017?

Response: Revised.

L.413-414, (Chen et al.,2016), there is no information in the reference list.

Response: Revised.

Reviewer 3 Report

Cao, et al. reviewed profiling miRNAs of teleost fish in response to environmental stress. The authors reviewed the expressions and functions of miRNAs and their targets identified under various stress conditions. The topic of the review is unique and potentially helpful to this field of study. The review itself is informative and provides a broad vision of miRNAs in different environmental stress. However, a few items need to be addressed for publication in biology.

Major Points:

-          Overall, the review provides a detailed explanation of multiple miRNAs in various fish species. However, it needs to be organized differently. The authors did a good job categorizing the context by the type of environmental stress, however, within each subcategory (type of environmental stress), there is an overwhelming amount of information in terms of fishes, miRNAs, and their functions. There are too many paragraphs under each subcategory.  I think that the authors need to add more subcategories under each type of stress. For example, 2. Salinity,    2.1. low salinity.. 2.2. high salinity.. or 2.1. freshwater..  2.2 brackish water.. 2.3. salt water.., etc. or even categorize by its targets or tissue types. These are just some suggested examples however better organization will make it easier for the readers to understand.

-          The authors need to separate the topics: changes of (expression or levels of) miRNAs in response to the environmental changes vs miRNA functions and its target mRNAs under the specific environment. Many examples are biological functions of the ‘miRNA targets’ under certain stress conditions without regulation of the miRNA in response to the environmental stress.

-          The authors also need to explore the regulatory mechanisms for how the environment causes changes in the miRNA levels. For example, the authors briefly stated that epigenetic changes can be linked through miRNAs in session 8, however, the authors need to address and explain in detail how the stress can cause the change of miRNAs and/or post transcriptional silencing. This study should be well-characterized in zebrafish.

-          The authors also need to address if the environmental changes/stress can influence any miRNA mediated silencing machinery, not only miRNA levels. For examples, are there any changes in Dicer, AGO1, or other RISC components under these stress conditions?

Minor Points:

-          Usage of the space: Overall, the figures are too large, and some of them can be combined. Figure 1 has very little information and can be removed. Figures 2,3,4, and 7 can be combined into one figure. The size of figures 5 and 6 is too large.

-          Is there any study on starvation conditions, especially linked to autophagy related miRNAs? If so, this can be classified under feed conditions.

Author Response

Comments and Suggestions for Authors

Cao, et al. reviewed profiling miRNAs of teleost fish in response to environmental stress. The authors reviewed the expressions and functions of miRNAs and their targets identified under various stress conditions. The topic of the review is unique and potentially helpful to this field of study. The review itself is informative and provides a broad vision of miRNAs in different environmental stress. However, a few items need to be addressed for publication in biology.

Many thanks for your positive comments and good advice.

Major Points:

-          Q1) Overall, the review provides a detailed explanation of multiple miRNAs in various fish species. However, it needs to be organized differently. The authors did a good job categorizing the context by the type of environmental stress, however, within each subcategory (type of environmental stress), there is an overwhelming amount of information in terms of fishes, miRNAs, and their functions. There are too many paragraphs under each subcategory.  I think that the authors need to add more subcategories under each type of stress. For example, 2. Salinity,    2.1. low salinity.. 2.2. high salinity.. or 2.1. freshwater..  2.2 brackish water.. 2.3. salt water.., etc. or even categorize by its targets or tissue types. These are just some suggested examples however better organization will make it easier for the readers to understand.

Response:

Many thanks for your good advice. Most miRNA can function in different environments: miRNA-122, miRNA-190b, miRNA-124, miRNA-1a and miRNA-206 were described in gills of marbled eels (Anguilla marmorata) adapted to different salinities (freshwater, brackish water and seawater) (Wang et al. 2015). Even though some miRNAs act in specific hyper-salinity environments, this has not been demonstrated in low-salinity environments. For example, miRNA-30c can work on renal HSP70 expression under hyper-osmoregulation but its role was not reported in hypo-osmoregulation (Yan et al. 2012), etc. Special miRNA seldom plays roles in the single salinity environment. We try to reorganize more miRNAs majoring in different environment instead adding subcategories under each type of stress.

Wang, X., Yin, D., Li, P., Yin, S., Wang, L., Jia, Y. and Shu, X. (2015) Microrna-sequence profiling reveals novel osmo-regulatory microrna expression patterns in catadromous eel Anguilla marmorata. PLoS One 10(8), e0136383.

Yan, B., Guo, J.T., Zhao, L.H. and Zhao, J.L. (2012) MiR-30c: a novel regulator of salt tolerance in tilapia. Biochem Bi-ophys Res Commun 425(2), 315-320.

-          Q2) The authors need to separate the topics: changes of (expression or levels of) miRNAs in response to the environmental changes vs miRNA functions and its target mRNAs under the specific environment. Many examples are biological functions of the ‘miRNA targets’ under certain stress conditions without regulation of the miRNA in response to the environmental stress.

Response:

Yes, miRNA expression occurs in response to the environmental changes in one condition and miRNA can play roles in some biological regulation under specific environment. We separate the two topics and some parts or paragraphs are reorganized. Thanks for your good advice.

-         Q3) The authors also need to explore the regulatory mechanisms for how the environment causes changes in the miRNA levels. For example, the authors briefly stated that epigenetic changes can be linked through miRNAs in session 8, however, the authors need to address and explain in detail how the stress can cause the change of miRNAs and/or post transcriptional silencing. This study should be well-characterized in zebrafish.

Response:

Thanks for your good comment. We cited some examples to state the regulatory mechanisms for how the environment causes changes in the miRNA levels about post transcriptional silencing.

Revise in corresponding parts:

miRNAs can silence the specific gene expression in complementary mRNAs from the translational repression and decay (Kobayashi and Singer 2022). Splicing factor proline protein can regulate miRNA silencing on specific binding sites and globally impact miRNA-dependent gene expression programs (Bottini et al. 2017). Many studies also reported that miRNAs can provide reversible gene silencing mechanisms in aestivation and hibernation and involved in cellular adaptation to specific demands under stressful conditions (Jones-Rhoades and Bartel 2004; Chang et al. 2022). In recent years, epigenetic factors have been thought to overwhelm gene expression and corresponding protein products. For example, epigenetic and energetically costly transcription or translation repression support metabolic suppression in chronically hypoxic goldfish. Goldfish can rely on transcriptional silencing of chronically hypoxic brain and heart via hypermethylation after being transiently hypomethylated, which suggested that transcriptional modifications support metabolic suppression but require a long hypoxia exposure to occur in these critical tissues (Farhat et al. 2022). miRNA is involved in post-transcriptional genes but its mechanisms in disease conditions are still largely unclear. Each miRNA has several hundred predicted target miRNAs, but only a small number of interactions have been experimentally validated.

Kobayashi, H., & Singer, R. H. (2022). Single-molecule imaging of microRNA-mediated gene silencing in cells. Nature communications, 13(1), 1-14.

Bottini, S., Hamouda-Tekaya, N., Mategot, R., Zaragosi, L. E., Audebert, S., Pisano, S., ... & Trabucchi, M. (2017). Post-transcriptional gene silencing mediated by microRNAs is controlled by nucleoplasmic Sfpq. Nature communications, 8(1), 1-16.

Chang, M., Li, B., Liao, M., Rong, X., Wang, Y., Wang, J., ... & Wang, C. (2022). Differential expression of miRNAs in the body wall of the sea cucumber Apostichopus japonicus under heat stress. Frontiers in Physiology, 1442.

Jones-Rhoades, M. W., & Bartel, D. P. (2004). Computational identification of plant microRNAs and their targets, including a stress-induced miRNA. Molecular Cell. 14, 787–799

Farhat, E., Talarico, G. G., Grégoire, M., Weber, J. M., & Mennigen, J. A. (2022). Epigenetic and post-transcriptional repression support metabolic suppression in chronically hypoxic goldfish. Scientific reports, 12(1), 1-16.

-          Q4) The authors also need to address if the environmental changes/stress can influence any miRNA mediated silencing machinery, not only miRNA levels. For examples, are there any changes in Dicer, AGO1, or other RISC components under these stress conditions?

Response:

Thanks for your good advice. In fact, we added a new paragraph to state the miRNA mediated silencing machinery in corresponding parts:

Very few genetic studies stated that environmental stress can influence any miRNA mediated silencing machinery in fish, demonstrating a major gap in the scientific literature in this field compared with mammal biology. Dicer endoribonucleases generate small RNAs for miRNA and RNA interference pathways (Paturi and Deshmukh 2021). Mammalian dicer supported the essential gene-regulating miRNA pathway, but how it is specifically adapted to miRNA biogenesis is unknown (Zapletal et al. 2022). Mature miRNA in the cytoplasm are known to be associated with Argonaute proteins (AGOs) in the RNA induced silencing complex (RISC), the RISC forms a complex with the miRNA by guiding the effector complex to the target mRNA for gene regulation (Macfarlane and Murphy 2010; Müller et al. 2022). Recent progress of dicer-related research, mature miRNA enriched in the nucleus associated with RISC proteins and potential RNA interference pathways will be discussed in aquatic species.

Paturi, S., and Deshmukh, M.V. (2021). A glimpse of "dicer biology" Through the structural and functional perspective. Front. Mol. Biosci. 8, 643657.

Zapletal, D., Taborska, E., Pasulka, J., Malik, R., Kubicek, K., Zanova, M., ... & Svoboda, P. (2022). Structural and functional basis of mammalian microRNA biogenesis by Dicer. Molecular Cell, 82(21), 4064-4079.

Müller, M., Fäh, T., Schaefer, M., Hermes, V., Luitz, J., Stalder, P., ... & Ciaudo, C. (2022). AGO1 regulates pericentromeric regions in mouse embryonic stem cells. Life science alliance, 5(6).

Minor Points:

-          Usage of the space: Overall, the figures are too large, and some of them can be combined. Figure 1 has very little information and can be removed. Figures 2,3,4, and 7 can be combined into one figure. The size of figures 5 and 6 is too large.

Response:

We the authors tried to combine the pictures, but found that the typesetting effect was not good. Please understand we keep the original idea presenting each picture separately. Many thanks for your advice.

-          Is there any study on starvation conditions, especially linked to autophagy related miRNAs? If so, this can be classified under feed conditions.

Response:

Added in corresponding parts:

Starvation can induce autophagy and disrupt antioxidant systems. Autophagy-related genes were activated and starvation disrupts intestinal homeostasis in the Chinese perch (Pan et al. 2022). miR-252 increased autophagy and antioxidant enzyme activity by targeting PI3K to enhance the tolerance of Penaeus vannamei in ammonia nitrogen stress (Wang et al. 2021). While few miRNA studies stated about the autophagy when fish starved. Future miRNA studies can focus on this area.

Pan, Y., Tao, J., Zhou, J., Cheng, J., Chen, Y., Xiang, J., ... & Chu, W. (2022). Effect of starvation on the antioxidative pathway, autophagy, and mitochondrial function in the intestine of Chinese perch Siniperca chuatsi. Aquaculture, 548, 737683.

Wang, F., Huang, L., Liao, M., Dong, W., Liu, C., Zhuang, X., ... & Wang, W. (2021). Pva-miR-252 participates in ammonia nitrogen-induced oxidative stress by modulating autophagy in Penaeus vannamei. Ecotoxicology and Environmental Safety, 225, 112774.

Round 2

Reviewer 1 Report

The manuscript has improved; however, the authors should mark in red the references that were checked and any text that was corrected.  They should also provide a detailed list of these changes. 

Table 1. convert the column “type” to rows showing each type of stress.

Unfortunately, the figure legends do not clearly explain the corresponding figure. Therefore, please add a figure legend to each image, instead of simply listing its components.

Homogenize the font size in all figures and also correct the typographical spaces in the figures.

I don't see any reason for conserving figure 5. It should be combined with another figure and have its relevance clarified.

Please check that references conform to the journal format.

Author Response

Reviewer 1

The manuscript has improved; however, the authors should mark in red the references that were checked and any text that was corrected. They should also provide a detailed list of these changes. 

Response: Many thanks for your kind comments. We rechecked the references and figure legends.

Table 1. convert the column “type” to rows showing each type of stress.

Response: Added.

Unfortunately, the figure legends do not clearly explain the corresponding figure. Therefore, please add a figure legend to each image, instead of simply listing its components.

Response: Revised.

Homogenize the font size in all figures and also correct the typographical spaces in the figures.

Response: Revised.

I don't see any reason for conserving figure 5. It should be combined with another figure and have its relevance clarified.

 Response: Deleted.

Please check that references conform to the journal format.

Response: Well checked.

Reviewer 2 Report

There are a few citation errors (e.g. in table 1, Yan et al., 2012 -> 2013? Vasadia et al., 2019 -> can’t find it in Reference list) in the revised manuscript, but most of my points have been improved. After re-checking citation more carefully, the manuscript will be accepted.

Author Response

Reviewer 2

There are a few citation errors (e.g. in table 1, Yan et al., 2012 -> 2013? Vasadia et al., 2019 -> can’t find it in Reference list) in the revised manuscript, but most of my points have been improved. After re-checking citation more carefully, the manuscript will be accepted.

Response: Well checked.
